# Fourier Sensitivity and Regularization of Computer Vision Models

**Kiran Krishnamachari**                                                          *kirank@u.nus.edu*
*Institute for Infocomm Research, A\*STAR, Singapore*
*School of Computing, National University of Singapore, Singapore*

**See-Kiong Ng**                                                                  *seekiong@nus.edu.sg*
*Institute of Data Science, National University of Singapore, Singapore*
*School of Computing, National University of Singapore, Singapore*

**Chuan-Sheng Foo**                                               *foo__chuan__sheng@i2r.a-star.edu.sg*
*Institute for Infocomm Research, A\*STAR, Singapore*
*Centre for Frontier AI Research, A\*STAR, Singapore*

**Reviewed on OpenReview:** *https://openreview.net/forum?id=VmTYgjYloM*

## Abstract

Recent work has empirically shown that deep neural networks latch on to the Fourier statistics of training data and show increased sensitivity to Fourier-basis directions in the input. Understanding and modifying this Fourier-sensitivity of computer vision models may help improve their robustness. Hence, in this paper we study the frequency sensitivity characteristics of deep neural networks using a principled approach. We first propose a ***basis trick***, proving that unitary transformations of the input-gradient of a function can be used to compute its gradient in the basis induced by the transformation. Using this result, we propose a general measure of any differentiable model's ***Fourier-sensitivity*** using the unitary Fourier-transform of its input-gradient. When applied to deep neural networks, we find that computer vision models are consistently sensitive to particular frequencies dependent on the dataset, training method and architecture. Based on this measure, we further propose a ***Fourier-regularization*** framework to modify the Fourier-sensitivities and frequency bias of models. Using our proposed regularizer-family, we demonstrate that deep neural networks obtain improved classification accuracy on robustness evaluations.

## 1 Introduction

While deep neural networks (DNN) achieve remarkable performance on many challenging image classification tasks, they can suffer significant drops in performance when evaluated on out-of-distribution (o.o.d.) data. Intriguingly, this lack of robustness has been partially attributed to the frequency characteristics of data shifts at test time in relation to the frequency sensitivity characteristics of the model (Yin et al., 2019; Jo & Bengio, 2017). It is known that distinct spatial frequencies in images contain features at different spatial scales: low spatial frequencies (LSF) carry global structure and shape information whereas high spatial frequencies (HSF) carry local information such as edges and borders of objects (Kauffmann et al., 2014). Moreover, spatial frequencies may also differentially processed in the brain's visual cortex to learn features at different scales (Appendix A). We find that when information in frequencies that a model relies on is corrupted or destroyed, performance can suffer. Hence, understanding the frequency sensitivity of a DNN can help us characterise and improve them.

DNNs have been demonstrated to be sensitive to Fourier-basis directions in the input (Tsuzuku & Sato, 2019; Yin et al., 2019) both empirically and using theoretical analysis of linear convolutional networks

(Tsuzuku & Sato, 2019). In fact, the existence of so-called "universal adversarial perturbations" (Moosavi-Dezfooli et al., 2017), simple semantics-preserving distortions that can degrade models' accuracy across inputs and architectures, is attributed to this structural sensitivity. Yin et al. (2019) also showed that many natural and digital image corruptions that degrade model performance may also be targeting this vulnerability. Hence, understanding and modifying Fourier-sensitivity is a promising approach to improve model robustness. While this problem has been studied empirically, the precise definition and measurement of a computer vision model's *Fourier-sensitivity* still lacks a rigorous approach across studies. In addition, no principled method has been proposed to study and modify the Fourier-sensitivity of a model. Existing works have applied heuristic filters on convolution layer parameters (Wang et al., 2020; Saikia et al., 2021) and input data augmentations (Yin et al., 2019) to modify a model's frequency sensitivity.

In this work, we first propose a novel *basis trick*, proving that unitary transformations of a function's gradient can be used to compute its gradient in the basis induced by the transformation. Using this result, we propose a novel and rigorous measure of a DNN's Fourier-sensitivity using its input-gradient represented in the Fourier-basis. We demonstrate that DNNs are consistently sensitive to particular frequencies that are dependent on dataset, training method and architecture. This observation confirms that DNNs tend to rely on some frequencies more than others, which has implications for robustness when Fourier-statistics change at test time. Further, using our proposed measure, which is differentiable with respect to model parameters, we propose a framework of Fourier-regularization to directly modify the Fourier-sensitivities and frequency bias of a model. We show in extensive empirical evaluations that Fourier-regularization can indeed modify frequency characteristics of computer vision models, and can improve the generalization performance of models on o.o.d. datasets where the Fourier-statistics are shifted. In summary, our main contributions are:

1. We propose a **basis trick**, proving that unitary transformations of the input-gradient of any function can be used to compute its gradient in the basis induced by the transformation

2. We propose a novel and rigorous measure of a model's **Fourier-sensitivity** based on the unitary Fourier-transform of its input-gradient. We empirically show that Fourier-sensitivity of a model is dependent on the dataset, training method and architecture

3. We propose a framework of **Fourier-regularization** to directly induce specific Fourier-sensitivities in a computer-vision model, which modifies the frequency bias of models and improves generalization performance on out-of-distribution data where Fourier-statistics are shifted

## 2 Related work

### 2.1 Frequency perspectives of robustness

Yin et al. (2019); Tsuzuku & Sato (2019) characterised the Fourier characteristics of trained CNNs using perturbation analysis of their test error under Fourier-basis noise. They showed that a naturally trained model is most sensitive to all but low frequencies whereas adversarially trained (Madry et al., 2018) models are sensitive to low-frequency noise. They further showed that these Fourier characteristics relate to model robustness on corruptions and noise, with models biased towards low frequencies performing better under high frequency noise and vice versa. Abello et al. (2021) took a different approach by measuring the impact of removing individual frequency components from the input using filters on accuracy, whereas Ortiz-Jimenez et al. (2020) computed the margin in input space along basis directions of the discrete cosine transform (DCT). Wang et al. (2020) made observations about the Fourier characteristics of CNNs in different training regimes including standard and adversarial training by evaluating accuracy on band-pass filtered data. Contrary to these empirical approaches, we propose a rigorous measure of a model's *Fourier-sensitivity*.

### 2.2 Modifying frequency sensitivity of models

Yin et al. (2019) observed that adversarial training (Madry et al., 2018) and Gaussian noise augmentation can induce a low-frequency sensitivity on some datasets. Wang et al. (2020) proposed smoothing convolution filter parameters to induce a low-frequency sensitivity in models. We note that such techniques can, in

principle, be undone by subsequent layers of a network. Shi et al. (2022) proposed similar techniques in the context of deep image priors applied to generative tasks. In addition, data augmentations such as Gaussian noise do not provide precise control over the Fourier-sensitivity of a model. In this work, we propose a *Fourier-regularization* framework to precisely modify the Fourier-sensitivity of any differentiable model.

### 2.3 Jacobian regularization

Methods that regularize the input-Jacobian of a model can be broadly classified into two categories: methods that minimize the norm of the input-Jacobian, and those that regularize its direction or directional derivatives at the input. Drucker & Le Cun (1991) proposed a method that penalized the norm of the input-Jacobian to improve generalization; more recently, this has been explored to improve robustness to adversarial perturbations (Ross & Doshi-Velez, 2018; Jakubovitz & Giryes, 2018; Hoffman et al., 2019). Simard et al. (1992) proposed "Tangent Prop", which minimized directional derivatives of classifiers in the direction of local input-transformations (e.g. rotations, translations; called "tangent vectors") to reduce sensitivity to such transformations. Czarnecki et al. (2017) proposed Sobolev training of neural networks to improve model distillation by matching the input-Jacobian of the original model. Regularizing the direction of the input-Jacobian has also been used to improve adversarial robustness (Chan et al., 2020). In the present work, we regularize frequency components in the input-gradient to improve performance on out-of-distribution tasks. As such, we are interested in modifying the input-gradient along certain directions instead of its total norm.

## 3 Proposed methods

**Preliminaries:** Consider an image classification task with input $x$, labels $y$, and standard cross-entropy loss function $\mathcal{L}_{CE}$. Let $f$ denote any differentiable model that outputs a scalar loss, $\mathcal{F}(\cdot)$ the unitary discrete Fourier transform (DFT), $\mathcal{F}^{-1}(\cdot)$ its inverse, and $\mathcal{F}^{-1^*}(\cdot)$ the adjoint of the inverse-Fourier transform, and let $x_f$ denote the Fourier-space representation of the input, i.e. $x_f = \mathcal{F}(x)$. We denote the input-gradient in the standard basis as $J_f(x)$, and $J_f(x_f)$ as the input-gradient with respect to the input in the Fourier-basis. Let $N$ be the height of input images (although not necessary, all images used in this work are square).

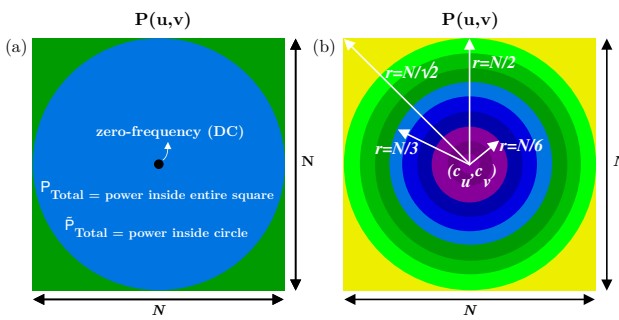

Figure 1: Power-matrix P of input-gradient.

**DFT notation:** The zero-shifted (rearrange DC component to centre and high frequencies further from center) 2D-DFT of the input-gradient is denoted $F$. Since the input-gradient typically has three color channels, they are averaged before computing the 2D-DFT. Fourier coefficients in $F$ are complex numbers with real and imaginary components; $F(u,v) = Real(u,v) + i \times Imag(u,v)$, where $(u,v)$ are indices of coefficients. The *power* in a coefficient is its squared amplitude i.e. $P(u,v) = |F(u,v)|^2 = Real(u,v)^2 + Imag(u,v)^2$ and the matrix of powers is denoted $P$ (power-matrix). Each coefficient has a radial distance $r(u,v)$ from the centre of the matrix, $r(u,v) = d((u,v),(c_u,c_v))$, where $(c_u,c_v)$ denotes the center of $P$ and $d(\cdot,\cdot)$ is Euclidean distance rounded to the nearest integer. Distinct radial distances of coefficients in the matrix are the set of integers $\{1,\dots,N/\sqrt{2}\}$ and correspond to low to high spatial frequencies, the highest frequency being limited by the Nyquist-frequency. We denote $P_{Total}$ as the total power in $P$, excluding the zero-frequency coefficient i.e. $P_{Total} = \sum_{r(u,v)>=1} P(u,v)$. Similarly, we define $\tilde{P}_{Total}$ as the total power in $P$ excluding the zero-frequency coefficient *and* coefficients with radial distance $r(u,v) > N/2$, i.e. coefficients outside the largest circle inscribed in $P$; $\tilde{P}_{Total} = \sum_{1<=r(u,v)<=N/2} P(u,v)$ (see Figure 1 for illustration). We denote $P_k$ as the power at radial distance $k$ normalized by $P_{Total}$, $P_k = \frac{1}{P_{Total}} \sum_{r(u,v)=k} P(u,v)$ and $\tilde{P}_k$ as the power at radial distance $k$ normalized by $\tilde{P}_{Total}$, $\tilde{P}_k = \frac{1}{\tilde{P}_{Total}} \sum_{r(u,v)=k} P(u,v)$.

### 3.1 *Basis Trick*: Unitary transformations of the input-gradient

In this section, we prove that unitary transformations of the input-gradient of a function provide its gradient in the new basis induced by the transformation. We term this the *basis trick* and use it to compute the Fourier-sensitivity of a model using the Fourier-transform of its input-gradient. To illustrate the *basis trick*, consider the computation graph in Figure 2 where the input $x$ in the standard basis is mapped to an output via a function $f$. We introduce an implicit operation (shown in red) that maps the Fourier-space representation of the input to the standard basis via the inverse Fourier-transform, i.e. $x_f \xrightarrow{\mathcal{F}^{-1}} x$. In order to compute the input-gradient with respect to input in the Fourier-basis, $J_f(x_f)$, we must differentiate through this *implicit* operation in the forward graph. Since the inverse-Fourier transform is a unitary operator, we have that $\mathcal{F}(J_f(x)) = J_f(x_f)$, due to the chain rule (see Corollary 1 below). Hence, even though we do not explicitly compute the Fourier-space representation of the input, this shows that the Fourier transform of the input-gradient provides the gradient of the model with respect to the input in Fourier-space. Analogous results can be obtained for other unitary operators such as the discrete cosine transform (DCT) and discrete wavelet transform (DWT) (see Proposition 1 below). In addition, this approach can be extended to $n$-dimensional input, e.g. time-series or 3D signals, by using the $n$-dimensional Fourier-transform. We formalize the *basis trick* below as a proposition and its corollary when the unitary operator is the Fourier-transform.

**Definition 1** (Unitary Operators). *A bounded linear operator $U : H \to H$ on a Hilbert space $H$ is said to be unitary if $U$ is bijective and its adjoint $U^* = U^{-1}$. Moreover, if $U$ is unitary, $U^{-1}$ is also a bounded and unitary linear operator.*

**Lemma 1** (Generalized Chain Rule). *Let $f$ be a scalar valued function of a vector $x$, and $A$ be a bijective linear operator such that $x = Ax_a$. Then, $A^*(J_f(x))$ is the gradient of $f$ with respect to $x_a$ i.e. $J_f(x_a) = A^*(J_f(x))$, where $A^*$ is the adjoint of $A$.*

**Proposition 1** (Basis Trick). *Let $f$ be a scalar valued function of a vector $x$, and $A$ be a bijective linear operator such that $x = Ax_a$. Then, the gradient vector of $f$ w.r.t $x_a$, $J_f(x_a) = A^{-1}(J_f(x))$ iff $A$ is unitary.*
*Proof.* Since $x = Ax_a$, $J_f(x_a) = A^*(J_f(x))$ due to Lemma 1. Since $A^* = A^{-1}$ iff $A$ is unitary (Definition 1), we have that $J_f(x_a) = A^{-1}(J_f(x))$ iff $A$ is unitary. $\square$

**Corollary 1** (Fourier Basis Trick). *If $A = \mathcal{F}^{-1}$, the unitary inverse-Fourier operator such that $x = \mathcal{F}^{-1}x_f$ with $x_f$ being the Fourier-basis representation of $x$, we have $J_f(x_f) = \mathcal{F}(J_f(x))$ where $\mathcal{F} = (\mathcal{F}^{-1})^{-1}$.*

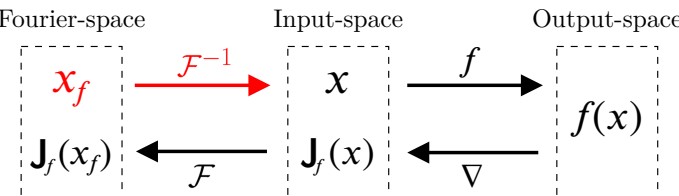

Figure 2: Fourier-transform of input-gradient is the gradient with respect to input in Fourier-space, i.e., $\mathcal{F}(J_f(x)) = J_f(x_f)$. Symbols in red represent the input in Fourier-space and need not explicitly computed.

### 3.2 Fourier-sensitivity of computer vision models

In this section, we define the ***Fourier-sensitivity*** of any differentiable model using its input-gradient represented in the Fourier-basis. Fourier-sensitivity is a measure of the relative magnitudes of a model's input-gradient with respect to different frequency bands in the input spectrum. As shown in Section 3.1, the input-gradient of a function with respect to the Fourier-basis can be computed by the unitary Fourier-transform of $J_f(x)$. To enable interpretation of the complete input-gradient in the Fourier-basis (see Appendix C.5 for examples), we summarize the information over frequency bands as shown in Figure 3. The Fourier-sensitivity $f_{SFS}(x, y)$ of a model with respect to an individual input $(x, y)$ is defined as

$$f_{SFS}(x, y) = [P_1, \ldots, P_{N/\sqrt{2}}] \tag{1}$$

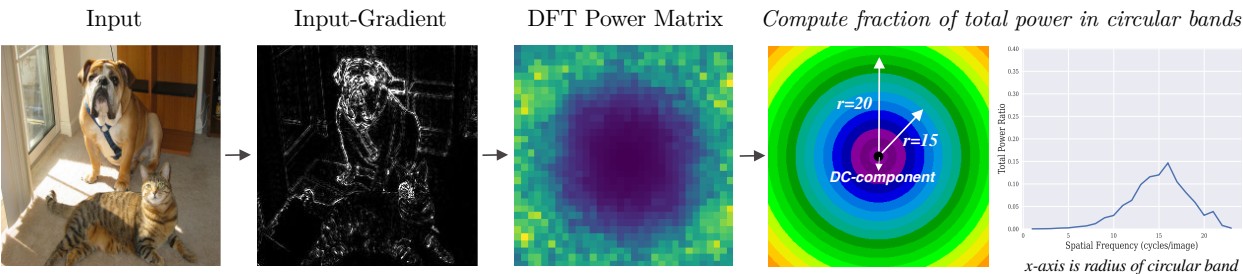

Figure 3: Computing *Fourier-sensitivity*. The input-gradient of the model is Fourier-transformed to obtain sensitivities with respect to frequencies. *Fourier-sensitivity* is then the vector with components being the proportion of total power in each circular frequency band.

where $P_k$ is the proportion of total power in Fourier coefficients at radial distance $k$ in the power matrix $P$ of $J_f(x_f)$. The overall *spatial frequency sensitivity (SFS)*, or simply *Fourier-sensitivity*, of a model is defined as the expectation of $f_{SFS}(x, y)$ over the data distribution $p(x, y)$, i.e. $f_{SFS}(\cdot; \theta) = \mathbb{E}_{(x,y)\sim p}[f_{SFS}(x, y)]$ (Algorithm 1 in Appendix B.1).

### 3.3 Fourier-regularization of computer vision models

In this section, we propose a framework of **Fourier-regularization**. Fourier-regularization enables control over the Fourier-sensitivity of a model by modifying the relative magnitude of a model's sensitivity to different frequency bands in the input spectrum. Fourier-regularization can modify the natural frequency sensitivity of neural networks as well as their generalization behavior. Our Fourier-regularizer augments the usual cross entropy loss: for a single example the new loss is $\mathcal{L}(x, y) = \mathcal{L}_{\mathrm{CE}}(x, y) + \lambda_{\mathrm{SFS}}\mathcal{L}_{\mathrm{SFS}}(x, y)$, where $\mathcal{L}_{\mathrm{SFS}}$ is the proposed regularizer and $\lambda_{\mathrm{SFS}}$ is a hyperparameter. Our regularizer penalizes the proportion of power in frequency bands based on the target Fourier-sensitivity. As $\mathcal{L}_{\mathrm{SFS}}$ is a function of the input-gradient, optimizing it requires an additional backpropagation step to compute derivatives with respect to parameters, similar to other gradient-regularization methods.

We now define $\mathcal{L}_{\mathrm{SFS}}$ for four instances of this regularizer: $SFS \in \{LSF, MSF, HSF, ASF\}$. Low-spatial-frequency (LSF) regularization trains a model to be insensitive to medium and high spatial frequencies, medium-spatial-frequency (MSF) regularization trains a model to be insensitive to low and high spatial frequencies, and high-spatial-frequency (HSF) regularization trains a model to be insensitive to low and medium spatial frequencies. These are achieved by penalizing the proportion of power, $P_k$, in the frequencies we wish the model to be insensitive to. All-spatial-frequency (ASF) regularization trains a model to be equally sensitive to all frequency bands. The motivation behind ASF regularization model is to encourage a model to be sensitive to multiple frequency bands instead of being concentrated in a small frequency range. Hence, the ASF-regularizer loss is defined as the negative entropy of the distribution of power over frequency bands. The definitions of *low*, *medium* and *high* frequency ranges are based on equally dividing the radius of the largest circle inscribed in the power-matrix $P$ into three equal parts (Figure 1b). For ASF-regularization, very high frequency bands, i.e. $r(u, v) > N/2$ are excluded, which is reflected in the $\tilde{P}_k$ terms. $\tilde{P}_k$ is the proportion of power in frequency bands within the largest circle inscribed in the power-matrix, P. Concretely, $\mathcal{L}_{\mathrm{SFS}}$ is defined for each of these three cases as follows:

| | LSF | MSF | HSF | ASF |
|---|---|---|---|---|
| $\mathcal{L}_{\mathrm{SFS}}$ | $\sum\limits_{k > N/6} P_k$ | $\sum\limits_{k < N/6, k > N/3} P_k$ | $\sum\limits_{k < N/3} P_k$ | $\sum\limits_{k=1}^{N/2} \tilde{P}_k \log \tilde{P}_k$ |

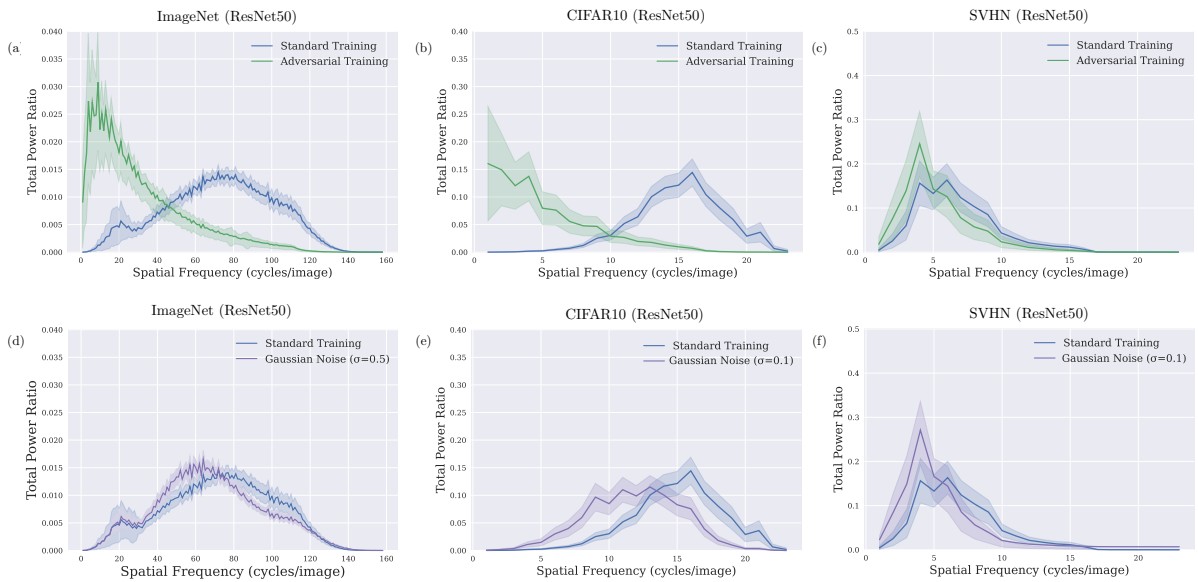

Figure 4: Fourier-sensitivity of (a),(b),(c) standard and adversarial training, (d),(e),(f) standard and Gaussian noise augmented training on ImageNet, CIFAR10 and SVHN with ResNet50 backbone.

## 4 Experiments

We first study below the Fourier-sensitivity of various architectures and training methods across datasets (Section 4.2). We found that both training method and architecture can have a significant impact on Fourier-sensitivity. We then identify an interesting connection between adversarial attacks and Fourier-sensitivity. Further, we study the effects of Fourier-regularization on representation learning (Section 4.3) as well as real o.o.d. benchmarks (Section 4.4).

### 4.1 Experimental setup

**Fourier-sensitivity analysis:** Fourier-sensitivity was computed by averaging across 1000 randomly selected validation samples for all datasets and shaded areas in plots represent two standard-deviations. We computed the Fourier-sensitivity of pre-trained ImageNet architectures obtained from *PyTorch Image Models* (Wightman, 2019). On CIFAR10 and CIFAR100 (Krizhevsky & Hinton, 2009), we trained all models for 150 epochs using stochastic gradient descent (SGD) with momentum (0.9), an initial learning rate of 0.1 decayed by a factor of 10 every 50 epochs, weight decay parameter equal to $5 \times 10^{-4}$ and batch size equal to 128. On SVHN (Netzer et al., 2011), we trained models for 40 epochs using Nesterov momentum with an initial learning rate of 0.01 and momentum parameter 0.9. The training batch size was 128, L2 regularization parameter was $5 \times 10^{-4}$ and learning rate was decayed at epochs 15 and 30 by a factor of 10. The following standard data augmentations – random-crop, random-horizontal-flip, random-rotation, and color-jitter – were used during training.

**Fourier-regularization experiments:** We demonstrate Fourier-regularization using ResNet50, Efficient-NetB0, MobileNetV2 and DenseNet architectures. To evaluate the proposed regularizer on high-resolution images, we also trained models on a subset of ImageNet derived from twenty five randomly chosen classes with images resized to $224 \times 224$ (ImageNet-subset). They were trained with SGD till convergence (lr=0.1, weight decay=$1 \times 10^{-4}$; lr mutliplied by 0.1 every 50 epochs); all models converged within 200 epochs. We trained Fourier-regularized models using $\lambda_{\text{SFS}} = 0.5$, which was set as the smallest value that achieved the target Fourier-sensitivity computed on validation samples independently of performance on target distribution data. We benchmarked against methods that have been proposed to modify the frequency sensitivity of models such as adversarial training and Gaussian noise augmentation to induce low-frequency sensitivity

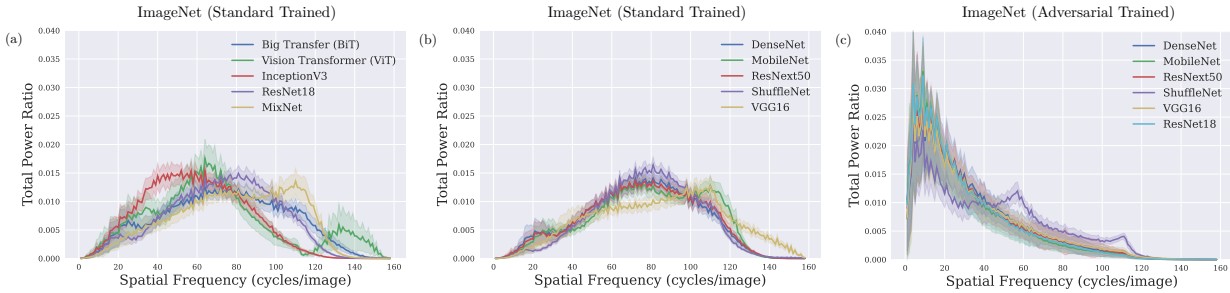

Figure 5: Fourier-sensitivities of multiple architectures after (a),(b) standard training and (c) adversarial training (PGD-$\ell_2$ ($\epsilon = 3$)) on ImageNet.

(Yin et al., 2019). For these methods, we used hyperparameter values most popular for training robust models in previous works. For adversarial training (AT), we used standard PGD $\ell_2$ attacks ($\epsilon = 1$ for CIFAR10/CIFAR100 and $\epsilon = 3$ for ImageNet-subset, attack-steps = 7 attack-lr = $\epsilon/7$). For Gaussian noise training, we added i.i.d. Gaussian noise drawn from $\mathcal{N}(0, \sigma^2)$ to each pixel during training ($\sigma = 0.1$). We used the *robustness* (Engstrom et al., 2019) library for training.

## 4.2 Fourier-sensitivity analysis

### 4.2.1 Fourier-sensitivity is dependent on dataset, training and architecture

We visualized the *Fourier-sensitivity* of models trained on ImageNet, SVHN and CIFAR10 (Figure 4; axes vary across datasets due to different image sizes). We observed that models are sensitive to some frequencies more than others and that this bias is consistent across samples (shaded areas represent two standard deviations across samples). Standard trained ImageNet models are in general sensitive to a wide range of the frequency spectrum with peak sensitivity to mid-range frequencies. The InceptionV3 (Szegedy et al., 2016) architecture is more sensitive to low-frequencies while Vision Transformer (ViT) (Dosovitskiy et al., 2021) displays sensitivity to mid-range as well as high frequencies (Figure 5a). In contrast, Big Transfer (BiT) (Kolesnikov et al., 2020), MixNet (Tan & Le, 2019b) and ResNet18 (He et al., 2016) models are sensitive to frequencies across the spectrum, with sensitivity tapering off at the high-frequencies (Figure 5a). These results suggest that model architecture can affect Fourier-sensitivity due to their different inductive biases. We further observed consistency of Fourier-sensitivity across popular convolutional architectures trained on ImageNet (Figure 5b). Adversarially trained models (Madry et al., 2018) are most sensitive to low spatial frequencies across datasets and architectures, which suggests that they rely on coarse global features as observed in prior work (Figures 4a, 4b, 4c, 5c). Gaussian noise augmented training slightly biases the model towards lower frequencies (Figures 4d, 4e, 4f, 10b) compared to baseline. Training on Stylized-ImageNet, proposed by Geirhos et al. (2019) to train shape-biased models, induces sensitivity to lower frequencies (Figure 11 in Appendix C.3), which reflects the increased shape-bias of these models.

Standard trained CIFAR10 models are most sensitive to high frequencies (Figure 4b), similar to CIFAR100 models (Figure 9 in Appendix C.1). In contrast, standard training on SVHN leads to a low-frequency sensitivity (Figure 4c), which suggests a dataset dependence of Fourier-sensitivity. Interestingly, we observed that models trained on common corruptions of CIFAR10 borrowed from (Hendrycks & Dietterich, 2019) display different Fourier-sensitivities to a model trained on clean CIFAR10 images (Appendix C.2). For example, models trained on images with severe noise corruptions (Gaussian, shot and speckle noise) display increased sensitivity to lower frequencies (Figure 10b), as did models trained on highly Gaussian-blurred, Glass-blurred, JPEG-compressed and pixelated images (Figure 10a, 10c). These changes reflect the shift in the Fourier-statistics of these corrupted images.

Finally, Fourier-regularization modifies the Fourier-sensitivity of models across datasets (Figure 6). LSF-REGULARIZED models are most sensitive to low-frequencies, MSF-REGULARIZED models are most sensitive

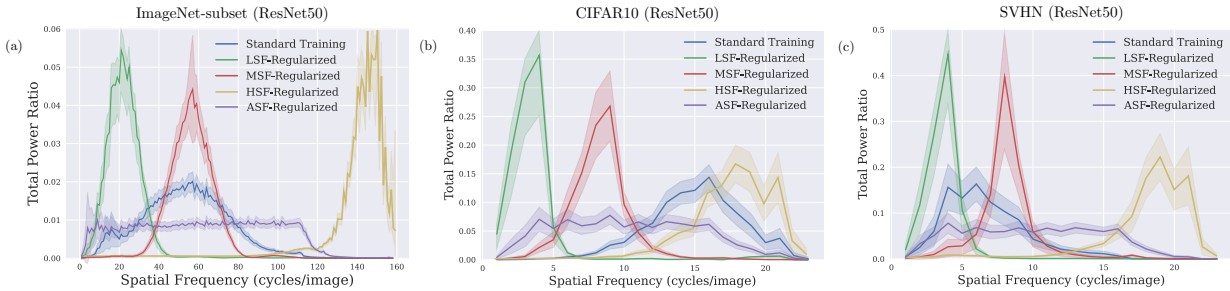

Figure 6: Fourier-sensitivity of models trained on (a) ImageNet-subset, (b) CIFAR10, (c) SVHN.

to the mid-frequency range, HSF-REGULARIZED models are most sensitive to high-frequencies, and ASF-REGULARIZED models are sensitive to a wide frequency range. Further, Fourier-regularization is demonstrated to be effective across architectures (Figure 12 in Appendix C.4).

### 4.2.2 Fourier-sensitivity and adversarial attacks

Adversarial attacks are imperceptible perturbations that can drastically reduce the classification performance of computer vision models. Many methods have been proposed to defend against as well as analyse the properties of such perturbations, including frequency-based approaches. Contrary to opinions that adversarial attacks are strictly a low-frequency or high-frequency phenomenon, we observed that adversarial perturbations closely resemble the target models' Fourier-sensitivity, which varies with dataset, training method and architecture as we have shown. This connection naturally arises from the fact that common gradient-based adversarial attack procedures such as PGD (Projected Gradient Descent) (Madry et al., 2018) typically use the direction of the input-gradient to generate perturbations. Plotting models' Fourier-sensitivities along with the Fourier power-spectra of adversarial perturbations shows this connection (Figure 14 in Appendix C.6). Consistent with observations made by Sharma et al. (2019) that adversarially trained ImageNet models are still vulnerable to low-frequency constrained perturbations in some settings, the Fourier-sensitivity of adversarially trained ImageNet models is concentrated in low-frequencies (Figures 4d, 4e, 4f, 5c). Sharma et al. (2019) also observed that low-frequency constrained attacks cannot easily fool standard trained ImageNet models, for which the Fourier-sensitivity has its peak at medium and high frequencies (Figure 4a, 4b) and are hence less vulnerable to low-frequency constrained attacks. We also observed that adversarial attacks against Fourier-regularized models have matching power-spectra (Figure 14b in Appendix C.6). For example, PGD adversarial perturbations against MSF-REGULARIZED models have power-spectra concentrated in mid-range frequencies. This suggests that adversarial perturbations are not a low or high-frequency phenomena but depend on the Fourier-sensitivity characteristics of a model.

### 4.3 Fourier-regularization modifies the frequency bias of models

Here we demonstrate that Fourier-regularization modifies the input frequencies that a model relies on using evaluations in different settings.

### 4.3.1 Validating Fourier-regularization using frequency-specific noise

We investigate the sensitivity of Fourier-regularized models to Fourier-basis directions in the input using data-agnostic corruptions, which have also been identified as a threat to model security (Yin et al., 2019; Tsuzuku & Sato, 2019). A Fourier-noise corruption is additive noise containing a single Fourier-mode (frequency). These corruptions are semantics-preserving but affect model performance and can be used to evaluate the sensitivity of a model to individual frequencies (see Figure 7 and Appendix E for examples). We added noise at all frequencies to the respective test sets of SVHN and CIFAR10 to evaluate the sensitivity of models. On CIFAR10, the standard trained model has the highest error at medium-to-high frequencies (Figure 17a in Appendix E). The standard trained SVHN model makes the most errors when low-to-medium frequency noise is added to the input (Figure 18a in Appendix E). This is in agreement with their respective

| Original | Fourier-Filtered | Patch-Shuffled | Original | High-Frequency | Med-Frequency | Low-Frequency |
|---|---|---|---|---|---|---|

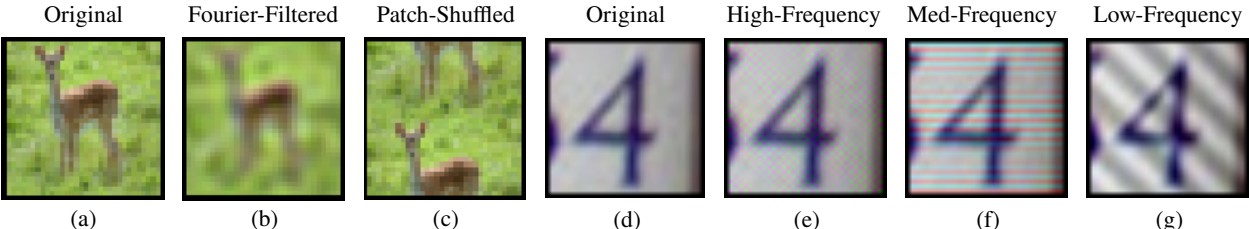

| (a) | (b) | (c) | (d) | (e) | (f) | (g) |

Figure 7: Examples (b): CIFAR10 Fourier-filtered (Section 4.3.3) (c): CIFAR10 Patch-shuffled (Section 4.3.2). (e) - (g): Fourier-noise corruptions on SVHN (Section 4.3.1). More examples in Appendix.

Fourier-sensitivities (Figure 4). Similarly, the LSF-REGULARIZED model is most sensitive to low-frequency perturbations and less so to medium and high-frequency distortions, across both CIFAR10 (Figure 17b) and SVHN (Figure 18b). The MSF-REGULARIZED models are most sensitive to mid-range frequencies (Figures 17c, 18c) and ASF-REGULARIZED models are sensitive to frequencies across the spectrum (Figures 17d, 18d). Detailed heat maps of error rates for noise across the all frequencies reflect the modified Fourier-sensitivities of Fourier-regularized models (Appendix E). This validates that the Fourier-regularization framework can indeed modify the sensitivity of models to frequencies in the input spectrum.

### 4.3.2    Learning global image features

As low frequency features correspond to large spatial scales while high frequency features are local in nature, Fourier-regularization allows us to bias the scale of features used by a model. Here we explore the extent to which Fourier-regularized models use global features by measuring their classification accuracy on patch-shuffled images, which have previously been used by Mummadi et al. (2021); Zhang & Zhu (2019); Wang et al. (2019). Patch-shuffling involves splitting an image into $k \times k$ squares and randomly swapping the positions of these squares. This is intended to destroy global features and retain local features; larger values of $k$ retain less global structure in the image (see Figure 7 and Appendix F for examples). As such, models that rely more on global rather than local structure suffer more from patch-shuffling. Hence, lower accuracy suggests increased reliance on global structure. We observed that LSF-REGULARIZED models, which are most sensitive to low-frequencies, as well as adversarially trained models, suffered large drops in accuracy, which suggests they rely on global structure in images (Table 1). This contrasts with standard trained, HSF-REGULARIZED and Gaussian noise augmented models, which retain higher accuracy under patch-shuffling. This reflects their bias towards learning local features instead of global structure on these datasets.

Table 1: Accuracy of ResNet50 on patch-shuffled CIFAR10 and CIFAR100 test sets.

| Method | CIFAR10 | | CIFAR100 | |
|---|---|---|---|---|
| | $k = 2$ | $k = 3$ | $k = 2$ | $k = 3$ |
| Std. Train | 66.5 | 45.8 | 39.9 | 21.4 |
| Gaussian Noise | 62.9 | 44.5 | 34.4 | 18.0 |
| HSF-REGULARIZED | 60.7 | 38.7 | 37.3 | 19.0 |
| LSF-REGULARIZED | **43.2** | **30.6** | 23.4 | 13.0 |
| MSF-REGULARIZED | 46.8 | 33.1 | 24.1 | 13.0 |
| ASF-REGULARIZED | 46.8 | 32.6 | 29.0 | 15.0 |
| AT (PGD $\ell_2, \epsilon = 1$) | 45.2 | 35.0 | **19.1** | **11.1** |

### 4.3.3   Robustness to Fourier-filtering

Jo & Bengio (2017) showed that DNNs have a tendency to rely on superficial Fourier-statistics of their training data. In the vein of generalization evaluations they performed, we generated semantics-preserving Fourier-filtered test images using radial masking in frequency space (see Figure 7 and Appendix D.1 for examples). A mask radius $r$ determines Fourier components that are preserved with larger radii preserving more components. We use $(c_u, c_v)$ to denote the centre of the mask and $d(\cdot, \cdot)$ to denote Euclidean distance. The mask is applied on the zero-shifted output of the Fourier transform of each image, denoted $X$, followed by the inverse transform, i.e. $X_{filtered} = \mathcal{F}^{-1}(\mathcal{F}(X) \odot M_r)$, where $\odot$ is the element-wise product. Formally, the radial mask is $M_r(u, v) := \begin{cases} 1, & \text{if } d((u,v),(c_u,c_v)) \leq r \\ 0, & \text{otherwise} \end{cases}$. Fourier-filtering is performed on each color channel independently. On ImageNet-scale images, LSF-REGULARIZATION is robust to significant low-pass filtering ($r = 37$), with just a $\sim 1\%$ drop in accuracy, whereas the baseline model drops by $\sim 7\%$ (Table 2). A standard trained CIFAR10 model suffers up to a 75% drop in accuracy on highly low-pass filtered data as it relies on high frequency information that is no longer present. On the other hand, other Fourier-regularized models perform robustly against Fourier-filtering. On CIFAR10, the LSF-REGULARIZED model performs robustly even on severely low-pass filtered images ($r = 5$), achieving an accuracy of 78.3% compared to the standard trained model's 18.6%. This shows that LSF-REGULARIZED CNNs are able to exploit low frequency features more than other models in the absence of high-frequency features. The adversarially trained (AT) model is significantly more robust than the baseline model due to its low-frequency sensitivity but not as robust as the LSF-REGULARIZED model. Gaussian noise augmentation does not provide significant robustness to Fourier-filtering. Both MSF-REGULARIZED and ASF-REGULARIZED models also provide significant robustness to Fourier-filtering while not as much as the LSF-REGULARIZED model. The HSF-REGULARIZED model was comparable to or more robust than the standard trained model. We further observed that other architectures – EfficientNetB0 (Tan & Le, 2019a), MobileNetV2 (Sandler et al., 2018) and DenseNet (Huang et al., 2017)– are also significantly vulnerable to Fourier-filtering. Fourier-regularization can similarly improve robustness over baseline methods on these architectures as well (Table 3, plots in Appendix C.4).

### 4.4   Fourier-regularization confers robustness to real out-of-distribution data shifts

Here we explore the robustness of Fourier-regularization on real o.o.d. data. Image corruptions in deployments of computer vision models can cause unfavorable shifts in the Fourier-statistics of data (Yin et al., 2019). For example, computer vision models deployed in vehicles may encounter motion blur due to movement, which can disrupt high-frequency information in images. Similarly, digital corruptions can cause similar effects on Fourier-statistics, such as JPEG compression artifacts and pixelation in low resolution settings. On ImageNet-C (Hendrycks & Dietterich, 2019), Fourier-regularization confers robustness to multiple corruptions. LSF-REG ($\lambda=1$) was most robust to blur corruptions, which carry most information in the low frequencies. ASF-REG ($\lambda=0.5$) provided significant robustness to weather and digital corruptions (Table 4), which suggests that sensitivity to a broad range of frequencies is needed to be robust to these corruptions.

Table 2: Accuracy of ResNet50 on Fourier-filtered ImageNet-subset, CIFAR10 and CIFAR100 test sets.

| Method | ImageNet-subset | | | CIFAR10 | | | | CIFAR100 | | | |
|---|---|---|---|---|---|---|---|---|---|---|---|
| | clean | $r = 37$ | $r = 20$ | clean | $r = 11$ | $r = 7$ | $r = 5$ | clean | $r = 11$ | $r = 7$ | $r = 5$ |
| Std. Train | 84.6 | 77.2 | 54.2 | 94.9 | 78.1 | 24.9 | 18.6 | 76.2 | 49.7 | 14.1 | 6.6 |
| LSF-REGULARIZED | 84.4 | **83.0** | **67.8** | 87.1 | 86.2 | **84.4** | **78.3** | 62.5 | 61.5 | **58.0** | **46.8** |
| MSF-REGULARIZED | 86.2 | 74.0 | 59.0 | 90.6 | **86.3** | 71.5 | 46.2 | 70.7 | **62.2** | 46.4 | 18.6 |
| HSF-REGULARIZED | 87.3 | 78.2 | 52.2 | 93.5 | 76.4 | 34.5 | 25.1 | 75.8 | 50.2 | 18.2 | 9.8 |
| ASF-REGULARIZED | 88.5 | 82.4 | 65.3 | 87.9 | 85.0 | 69.3 | 45.0 | 67.0 | 62.1 | 41.1 | 19.8 |
| AT-PGD | 81.8 | 75.8 | 54.3 | 81.6 | 80.2 | 76.1 | 67.5 | 58.8 | 56.8 | 50.0 | 40.2 |
| Gaussian-noise | 84.8 | 74.6 | 36.7 | 94.5 | 84.4 | 32.4 | 19.5 | 73.1 | 61.9 | 27.7 | 11.6 |

The HSF-REG ($\lambda$=0.5) model was more robust to weather and digital corruptions compared to blurring, which requires a low-frequency bias. Hence, modifying the Fourier-sensitivity can improve robustness under multiple o.o.d. shifts that affect model robustness.

## 5 Discussion

### 5.1 Fourier-regularizer selection

Selecting the regularizer (i.e., LSF, MSF, HSF, ASF) that gives the best performance on a (shifted) target data distribution may be done using cross-validation if labeled data from the target distribution is available. Otherwise, since model or regularizer selection in the absence of labeled data from the target distribution is generally a hard and unsolved problem, we suggest choosing the Fourier-regularizer based on prior knowledge about the frequency bias of the learning task on the target distribution. For example, we have shown that on many common image corruptions such as various forms of blurring, LSF-REGULARIZED models can perform well due to the loss of high-frequency information under blurring (Section 4.4). This agrees with previous work that has analysed the spectra of corrupted images (Yin et al., 2019). As demonstrated in Section 4.3.2, LSF-REGULARIZED models are also more reliant on global features, which are generally robust to local changes in image texture (Geirhos et al., 2019). On high-resolution images, we showed that encouraging models to use more frequencies using ASF-REGULARIZATION can improve clean accuracy (Table 4).

#### 5.1.1 $\lambda_{\text{SFS}}$ hyper-parameter selection

Fourier-regularization requires choosing the frequency bias as well as the hyperparameter $\lambda_{\text{SFS}}$. We note that when $\lambda_{\text{SFS}} = 0$, Fourier-regularization is equivalent to standard training. Hence, very small values may not modify the frequency bias significantly. We found that a useful heuristic is to set the parameter as small as possible to achieve the target frequency bias, which can be measured by computing the Fourier-sensitivity of the model using training or validation samples independently of performance on the target distribution. Values larger than this can unnecessarily decrease clean accuracy further without improving accuracy on the target distribution. For LSF-REGULARIZATION on CIFAR10, we found that increasing $\lambda_{\text{SFS}}$ from 0 to 0.5 gradually nudges the model towards low-frequencies (Figure 20 in Appendix G). As we increased $\lambda_{\text{SFS}}$ further from 0.5 to 1, clean accuracy decreased further without modifying the Fourier-sensitivity (Table 7 in Appendix G). Strictly restricting the model to have a particular frequency bias using large values of $\lambda_{\text{SFS}}$ may overly constrain model capacity. This procedure can be performed to identify optimal $\lambda_{\text{SFS}}$ values even in the absence of labeled target distribution data.

### 5.2 Fourier-regularization and clean accuracy

Fourier-regularization can affect the frequencies utilized by models in a given dataset. The effect of Fourier-regularization on clean accuracy depends on the dataset *and* the chosen frequency range (e.g., LSF, HSF, ASF). On high-resolution ImageNet-scale images ($224 \times 224$) we observed that encouraging the model to use a wide range of frequencies using (ASF-REGULARIZATION) improved generalization performance over

Table 3: Evaluating Fourier-filtered CIFAR10 using other architectures.

| Method | EfficientNetB0 | | | | MobileNetV2 | | | | DenseNet | | | |
|---|---|---|---|---|---|---|---|---|---|---|---|---|
| | clean | $r=11$ | $r=7$ | $r=5$ | clean | $r=11$ | $r=7$ | $r=5$ | clean | $r=11$ | $r=7$ | $r=5$ |
| Std. Train | 89.9 | 76.1 | 30.2 | 24.0 | 92.6 | 74.5 | 27.6 | 18.3 | 94.0 | 69.6 | 19.3 | 16.4 |
| LSF-REGULARIZED | 84.1 | 83.5 | **80.2** | **68.3** | 81.7 | 81.5 | **78.3** | **67.4** | 86.2 | 85.7 | **80.7** | **69.0** |
| MSF-REGULARIZED | 88.7 | **86.6** | 55.5 | 31.2 | 89.0 | **87.6** | 68.7 | 38.7 | 90.6 | **89.6** | 72.5 | 38.5 |
| HSF-REGULARIZED | 90.5 | 73.3 | 28.6 | 19.2 | 90.3 | 83.2 | 52.2 | 36.1 | 92.9 | 80.5 | 35.0 | 23.6 |
| ASF-REGULARIZED | 89.5 | 77.4 | 30.6 | 24.1 | 79.0 | 73.0 | 44.9 | 28.6 | 88.2 | 85.5 | 69.7 | 46.7 |
| Gaussian-noise | 89.3 | 78.4 | 35.4 | 26.1 | 91.2 | 79.3 | 41.0 | 26.5 | 93.2 | 82.3 | 30.4 | 21.1 |
| AT-PGD | 72.3 | 71.5 | 68.4 | 63.1 | 82.0 | 80.2 | 75.9 | 67.0 | 81.9 | 80.9 | 76.3 | 67.9 |

Table 4: Accuracy of ResNet50 on clean and corrupted (severity 2) test set of ImageNet-subset.

| Method | Clean | Blur | | | | Weather | | | | Digital | | | |
| | | Defocus | Glass | Motion | Zoom | Snow | Frost | Fog | Brightness | Contrast | Elastic | Pixel | JPEG |
|---|---|---|---|---|---|---|---|---|---|---|---|---|---|
| Std. Train | 84.6 | 59.2 | 68.8 | 71.4 | 69.8 | 46.8 | 51.7 | 44.6 | 79.8 | 40.9 | 71.8 | 82.1 | 74.1 |
| LSF-REG ($\lambda$=1) | 83.8 | **71.5** | **77.6** | **76.6** | **76.4** | 54.2 | 58.6 | 46.7 | 79.6 | 46.6 | **75.3** | 83.7 | 75.5 |
| MSF-REG ($\lambda$=1) | 85.8 | 58.3 | 64.4 | 70.1 | 68.2 | 47.5 | 51.0 | 39.6 | 81.2 | 37.8 | 71.1 | 79.7 | 76.0 |
| HSF-REG ($\lambda$=1) | 86.9 | 56.3 | 65.0 | 67.6 | 63.9 | 48.1 | 53.7 | 41.8 | 80.5 | 38.1 | 69.9 | 81.1 | 78.2 |
| ASF-REG ($\lambda$=1) | 85.5 | 62.3 | 68.6 | 72.5 | 70.2 | 50.3 | 55.8 | 40.5 | 80.7 | 42.4 | 72.1 | 81.9 | 75.5 |
| LSF-REG ($\lambda$=0.5) | 84.4 | 63.4 | 73.0 | 72.7 | 70.6 | 52.1 | 56.2 | 43.5 | 80.5 | 42.4 | 74.2 | 82.2 | 75.9 |
| MSF-REG ($\lambda$=0.5) | 86.2 | 61.6 | 67.5 | 72.0 | 71.3 | 52.0 | 56.8 | 53.1 | 81.6 | 46.7 | 72.2 | 82.1 | 78.0 |
| HSF-REG ($\lambda$=0.5) | 87.3 | 60.4 | 68.9 | 71.0 | 70.6 | 52.6 | 58.6 | 51.1 | 83.3 | 49.8 | 72.3 | 83.6 | 77.8 |
| ASF-REG ($\lambda$=0.5) | **88.5** | 65.8 | 74.3 | 74.7 | 73.9 | **58.2** | **63.5** | **59.7** | **84.6** | **53.0** | 74.6 | **85.8** | 78.1 |
| Gaussian-noise | 84.8 | 38.8 | 55.4 | 61.8 | 57.6 | 47.4 | 51.0 | 35.0 | 81.4 | 27.1 | 67.2 | 78.1 | 71.4 |
| AT-PGD | 81.8 | 58.4 | 65.5 | 67.0 | 66.3 | 50.3 | 47.8 | 13.7 | 79.0 | 18.5 | 66.5 | 78.3 | **79.0** |

the baseline (88.5% vs 84.6%), as did HSF-REGULARIZATION (Table 2). On SVHN, an easier task, Fourier-regularization did not have a significant effect as baseline models already achieve high clean accuracies (~96%) (Table 6 in Appendix E.3). CIFAR10 and CIFAR100 are more challenging small image tasks that require high spatial frequencies (HSF) to maximise clean accuracy. Hence, the HSF-REGULARIZED model had high clean accuracy while we observed a drop in the clean accuracy of LSF,MSF,ASF regularized models, although they performed better on o.o.d. data. In summary, Fourier-regularization is a generic framework that can be used to improve performance in both i.i.d. and o.o.d. settings.

### 5.3 Fourier-regularization vs training on Fourier-filtered data

Here we contrast Fourier-regularization and training on Fourier-filtered data to modify the frequency bias of models. We note that Fourier-regularization cannot be replicated by training on Fourier-filtered data. Low-pass filtered images completely discard information in higher frequencies, which may not be desirable. Moreover, in natural images, the amount of energy in frequency bands falls off rapidly at high frequencies (Hyvärinen et al., 2009), hence, medium and high-pass filtered natural images typically appear completely empty to the human eye without additional contrast maximisation and are still not easily recognizable (see Figure 16 in Appendix D.2). Hence, training on medium-pass Fourier-filtered CIFAR10 achieved a clean accuracy of only ~33% whereas the MSF-REGULARIZED model's clean accuracy is ~90% (Table 5 in Appendix D.2). Similarly, training on high-pass filtered CIFAR10 training samples achieved only ~15% accuracy on clean test samples, while HSF-REGULARIZATION can achieve 93.5% (Table 2). Due to the energy statistics across frequency bands in natural images, training on Fourier-filtered data is not successful for all but the lowest frequency bands, where most of their energy resides. On the other hand, the Fourier-regularization framework allows controlling the sensitivity to each frequency band. In addition, we note that ASF-REGULARIZATION cannot be realized using Fourier-filtering alone.

## 6  Conclusion

We proposed a novel *basis trick* and proved that unitary transformations of a function's input-gradient can be used to compute its gradient in the basis induced by the transformation. Using this result, we proposed a novel and rigorous measure of the *Fourier-sensitivity* of any differentiable computer vision model. We explored Fourier-sensitivity of various models and showed that it depends on dataset, training and architecture. We further proposed a framework of *Fourier-regularization* that modifies the frequency bias of models and can improve robustness where Fourier-statistics of data have changed. We demonstrated that Fourier-regularization is effective on different image resolutions, datasets (Table 2) as well as architectures (Table 3). More broadly, Fourier-sensitivity and regularization can also be extended to other data modalities like audio and time-series, where Fourier analysis of machine learning models may also be useful. As Fourier-analysis is an important and fundamental toolkit, the analysis and control of machine learning models enabled by our work may prove to be valuable for learning tasks beyond those explored in this paper.

**Acknowledgments**

We thank Bryan Hooi and Wenyu Zhang for helpful discussions about the project as well as feedback on an early draft of the paper. Individual fellowship support for Kiran Krishnamachari was provided by the Agency for Science, Technology, and Research (A*STAR), Singapore.

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

# A    Spatial frequency channels in the brain

To further motivate the spatial frequency perspective of visual learning, we briefly describe some relevant findings from neuroscience and vision research. While the brain does not strictly perform a Fourier-analysis of visual scenes, there has been mounting evidence over decades for spatial frequency channels in the human visual system that are physiologically independent and selectively responsive to distinct spatial frequency bands (Campbell & Robson, 1964; 1968). It has been posited that the use of different spatial frequency channels are determined by the demands of a given visual task through an attention mechanism (Schyns & Oliva, 1999; Rotshtein et al., 2010; Julesz & Papathomas, 1984). Spatial frequency channels enable us to attend to different spatial scales in a scene at a fixed viewing distance, similar to the focus lens in a camera (Figure 8). Similarly, computer vision models develop a *Fourier-sensitivity* (Section 3.2) that is dependent on the dataset, architecture and method used to train them.

```
EFEFEFEFEFEFEFEFEFEFEFEFEFEFEFEFEFEFEFEFEFEF
EFEFEFEFEFEFEFEFEFEFEFEFEFEFEFEFEFEFEFEFEFEF
EFEFEFEFEFEFEFEFEFEFEFEFEFEFEFEFEFEFEFEFEFEF
EFEFEFEFEFEFEFEFEFEFEFEFEFEFEFEFEFEFEFEFEFEF
EFEFEFEFEFEFEFEFEFEFEFEFEFEFEFEFEFEFEFEFEFEF
EFEFEFEFEFEFEFEFEFEFEFEFEFEFEFEFEFEFEFEFEFEF
EFEFEFEFEFononononononononononononononoEFEFEFEFEF
EFEFEFEFEFononononononononononononononoEFEFEFEFEF
EFEFEFEFEFononononononononononononononoEFEFEFEFEF
EFEFEFEFEFononononononononononononononoEFEFEFEFEF
EFEFEFEFEFononononononononononononononoEFEFEFEFEF
EFEFEFEFEFEFEFEFFEononononononEFEFEFEFEFEFEFEFE
EFEFEFEFEFEFEFEFFEononononononEFEFEFEFEFEFEFEFE
EFEFEFEFEFEFEFEFFEononononononEFEFEFEFEFEFEFEFE
EFEFEFEFEFEFEFEFFEononononononEFEFEFEFEFEFEFEFE
EFEFEFEFEFEFEFEFFEononononononEFEFEFEFEFEFEFEFE
EFEFEFEFEFEFEFEFFEononononononEFEFEFEFEFEFEFEFE
EFEFEFEFEFEFEFEFFEononononononEFEFEFEFEFEFEFEFE
EFEFEFEFEFEFEFEFFEononononononEFEFEFEFEFEFEFEFE
EFEFEFEFEFEFEFEFFEononononononEFEFEFEFEFEFEFEFE
EFEFEFEFEFEFEFEFFEononononononEFEFEFEFEFEFEFEFE
EFEFEFEFEFEFEFEFFEononononononEFEFEFEFEFEFEFEFE
EFEFEFEFEFEFEFEFFEononononononEFEFEFEFEFEFEFEFE
EFEFEFEFEFEFEFEFFEononononononEFEFEFEFEFEFEFEFE
EFEFEFEFEFEFEFEFFEononononononEFEFEFEFEFEFEFEFE
EFEFEFEFEFEFEFEFEFEFEFEFEFEFEFEFEFEFEFEFEFEFE
EFEFEFEFEFEFEFEFEFEFEFEFEFEFEFEFEFEFEFEFEFEF
EFEFEFEFEFEFEFEFEFEFEFEFEFEFEFEFEFEFEFEFEFEF
EFEFEFEFEFEFEFEFEFEFEFEFEFEFEFEFEFEFEFEFEFEF
EFEFEFEFEFEFEFEFEFEFEFEFEFEFEFEFEFEFEFEFEFEF
```

Figure 8: Letters at multiple spatial scales. This image comprises the letters 'o', 'n', 'E' and 'F' at a small spatial scale (corresponds to high-frequency). The letter 'T' is also visible at a larger spatial scale (LSF) formed by the specific arrangement of the letters 'o', 'n'. Identifying these letters requires processing features at multiple scales, enabled by distinct spatial frequency channels in our visual cortex. Our ability to recognize only one of these scales at a time is evidence for the physiological independence of spatial frequency channels in the brain. Image based on Julesz & Papathomas (1984).

# B  Fourier-sensitivity

## B.1  Pseudo-code

---

**Algorithm 1** Fourier-sensitivity of a model

---

**Input:** Labeled samples $\mathbf{L} = \{(x_i, y_i)\}_{i=1}^{n}$; a model $f$ with trained parameters $\theta$

**Output:** Estimated Fourier-sensitivity of model, $f_{SFS}(\cdot; \theta)$

**for** $i = 1$ **to** $n$ **do**

    Compute loss $\mathcal{L}_{\mathrm{CE}}(f(x_i), y_i)$ {forward pass}

    Backpropagate $\mathcal{L}_{\mathrm{CE}}$ to obtain $\frac{\partial \mathcal{L}_{\mathrm{CE}}}{\partial x_i}$ {input-gradient, averaged across color channels}

    $\frac{\partial \mathcal{L}_{\mathrm{CE}}}{\partial x_{f_i}} = \mathcal{F}(\frac{\partial \mathcal{L}_{\mathrm{CE}}}{\partial x_i})$ {unitary 2D DFT of input-gradient}

    $f_{SFS}(x_i, y_i) = [P_k;$ for k=1 to N/$\sqrt{2}]$ {see Equation 1; excludes DC component}

**end for**

$f_{SFS}(\cdot; \theta) = \frac{1}{n} \sum_{i=1}^{n} f_{SFS}(x_i, y_i)$ {estimated Fourier sensitivity of model}

---

## C   Supplementary plots

### C.1   Fourier-sensitivities of ResNet50 models trained on CIFAR10 and CIFAR100

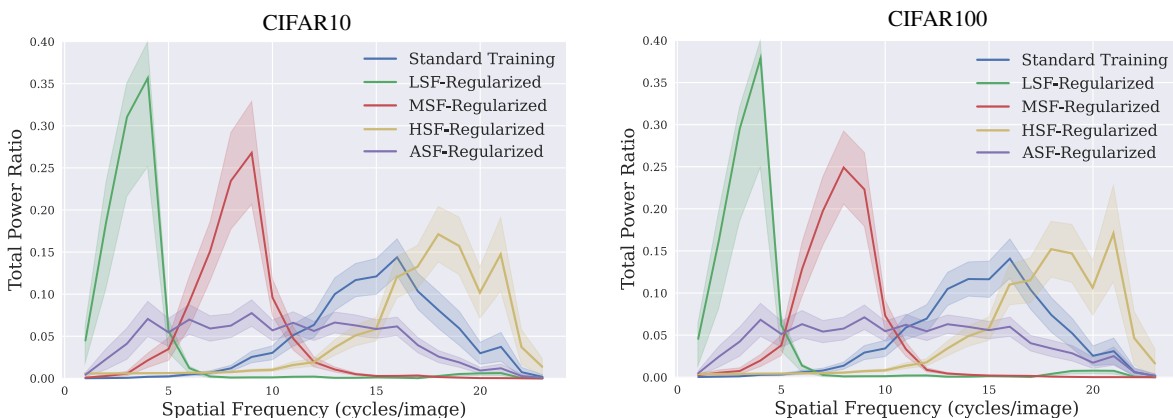

Figure 9: Fourier-sensitivity of models trained with various regularizers on CIFAR10 (left) and CIFAR100 (right). The shaded region represents two standard deviations.

### C.2   Training on corrupted CIFAR10

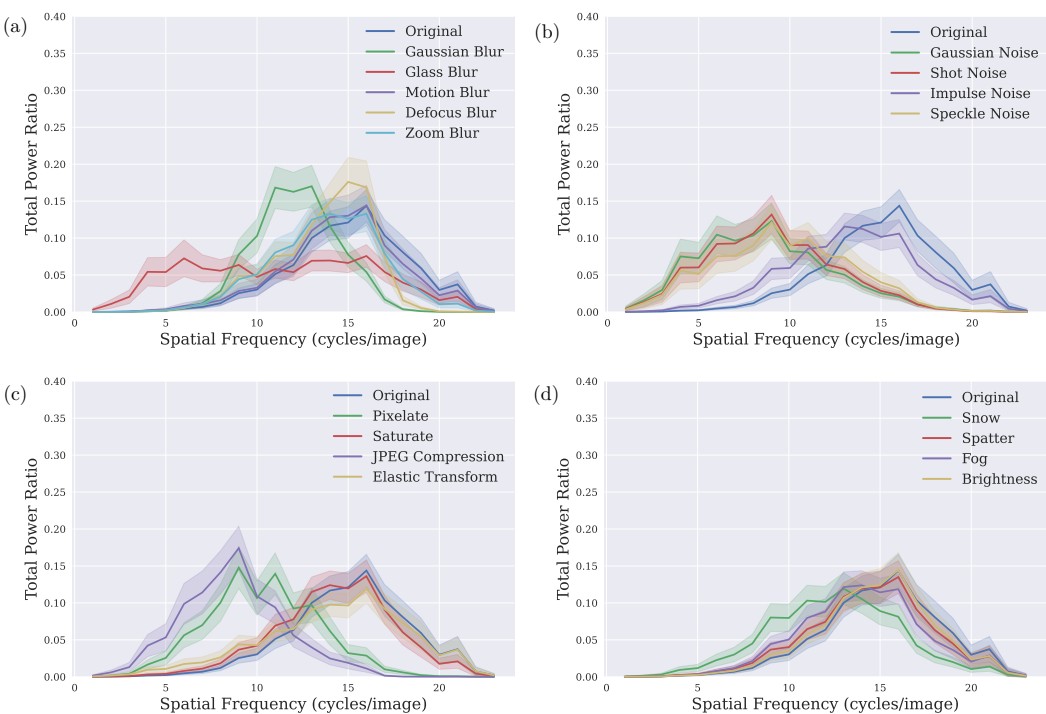

Figure 10: Fourier-sensitivity of ResNet50 models trained on the CIFAR10 training set distorted by corruptions derived from the CIFAR10-C (severity 5) dataset. The shaded region represents two standard deviations.

## C.3 Fourier-sensitivity of model trained on Stylized ImageNet

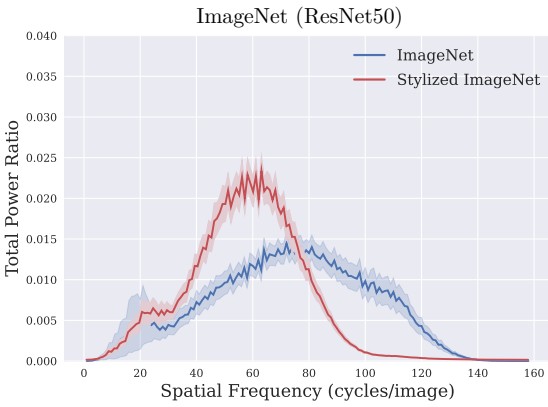

Figure 11: Fourier-sensitivity of ResNet50 trained on ImageNet and Stylized ImageNet.

## C.4 Fourier-regularization of other architectures

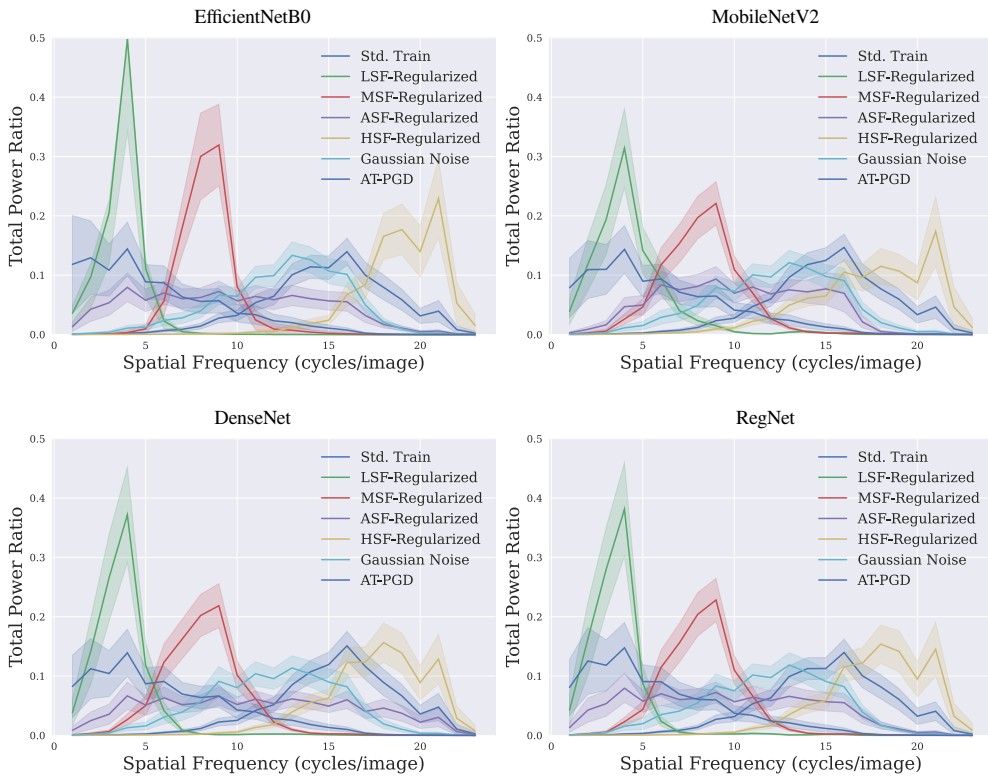

Figure 12: Fourier-sensitivity plots for other architectures (EfficientNetB0 (Tan & Le, 2019a), MobileNetV2 (Sandler et al., 2018), DenseNet (Huang et al., 2017), RegNet (Radosavovic et al., 2020)) trained on CIFAR10.

## C.5   Input-gradients in Fourier-basis of models trained on CIFAR10

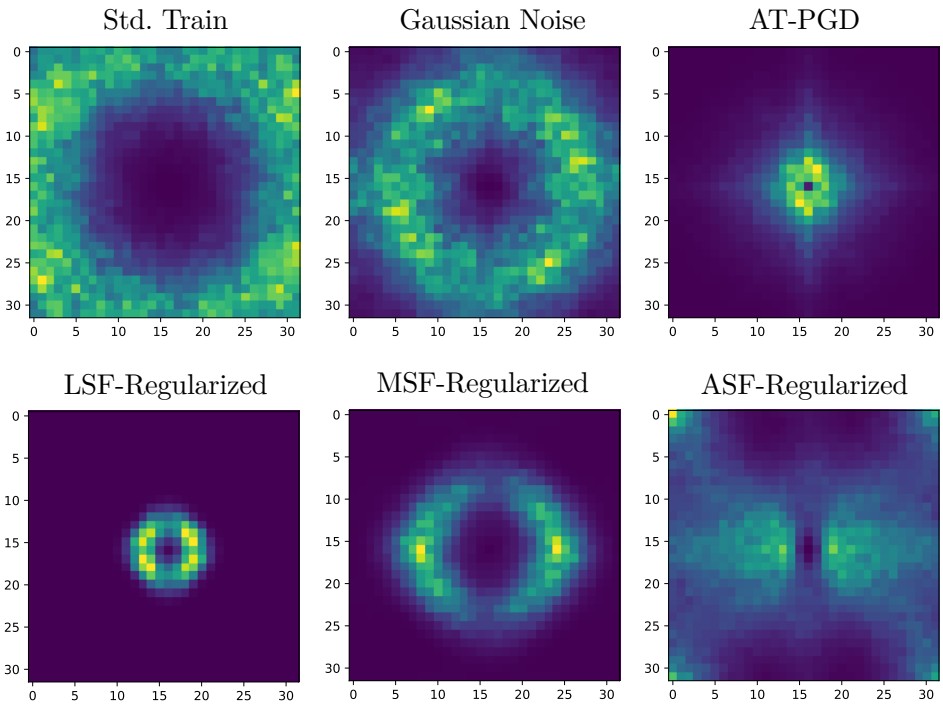

Figure 13: Input-gradients of ResNet50 models trained on CIFAR10, averaged across 1000 randomly chosen validation samples. Low frequencies are close to the center, high frequencies are further from the center.

## C.6 Fourier-sensitivity and adversarial attacks

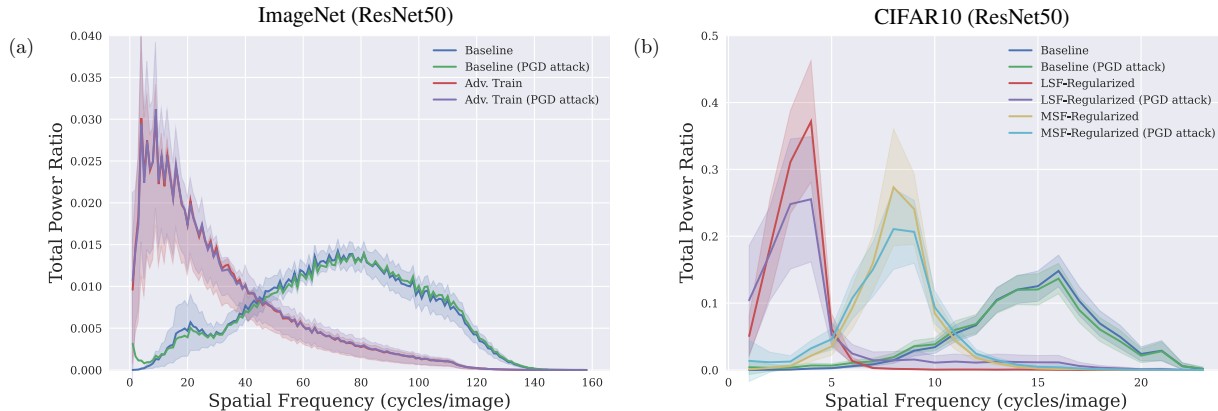

Figure 14: Power-spectra of adversarial perturbations align with Fourier-sensitivity of models. PGD $\ell_2$ attacks ($\epsilon = 3$) were used for each model.

# D Fourier-filtering

## D.1 Radial Fourier-filtering

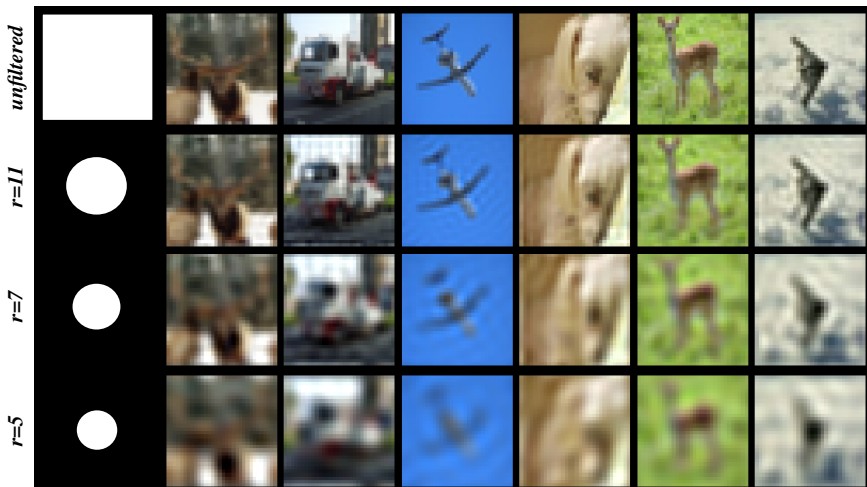

Figure 15: First image in each row is the mask in Fourier space (lowest frequency at centre). White pixels preserve and black pixels set Fourier components to zero. Top row are original CIFAR10 images, other rows are Fourier-filtered with different radial masks.

### D.2 Band-pass Fourier-filtering

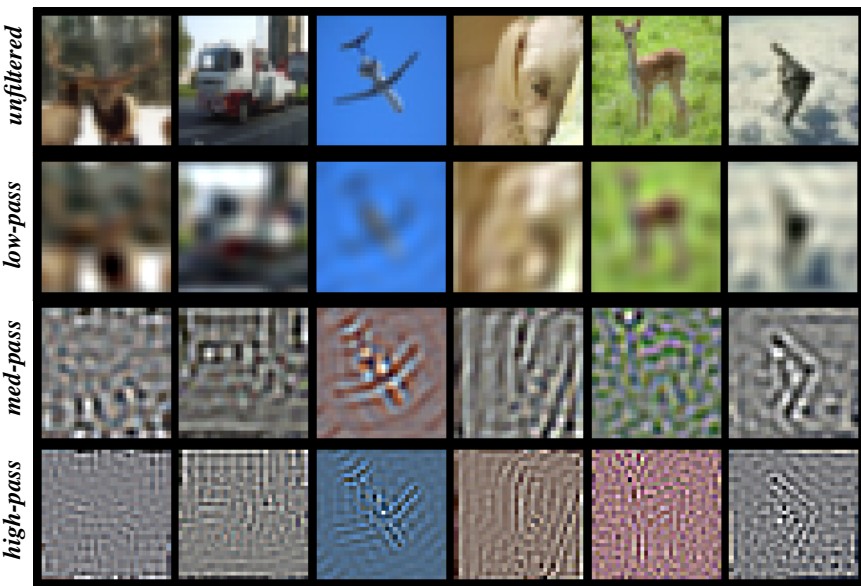

Figure 16: Band-pass Fourier-filtered CIFAR10 images. We filtered Fourier-coefficients in each color channel separately. For low-pass filtering, Fourier-coefficients with radial distance $r(u, v) > 5$ were set to zero. For medium-pass filtering, Fourier-coefficients with $r(u, v) < 5$ and $r(u, v) > 10$ were set to zero. For high-pass filtering, Fourier-coefficients with $r(u, v) < 10$ were set to zero. Medium-pass and high-pass filtered images were contrast-maximised for viewing.

Table 5: Comparing clean accuracy of models standard trained on filtered CIFAR10 images vs Fourier-regularization (ResNet50).

| Method | Accuracy |
|---|---|
| Low-pass Filtered | 86.6 |
| Medium-pass Filtered | 33.8 |
| High-pass Filtered | 15.3 |
| LSF–REGULARIZED | 87.1 |
| MSF–REGULARIZED | 90.6 |
| HSF–REGULARIZED | 93.5 |
| ASF–REGULARIZED | 87.9 |

# E Fourier-noise corruptions

## E.1 CIFAR10

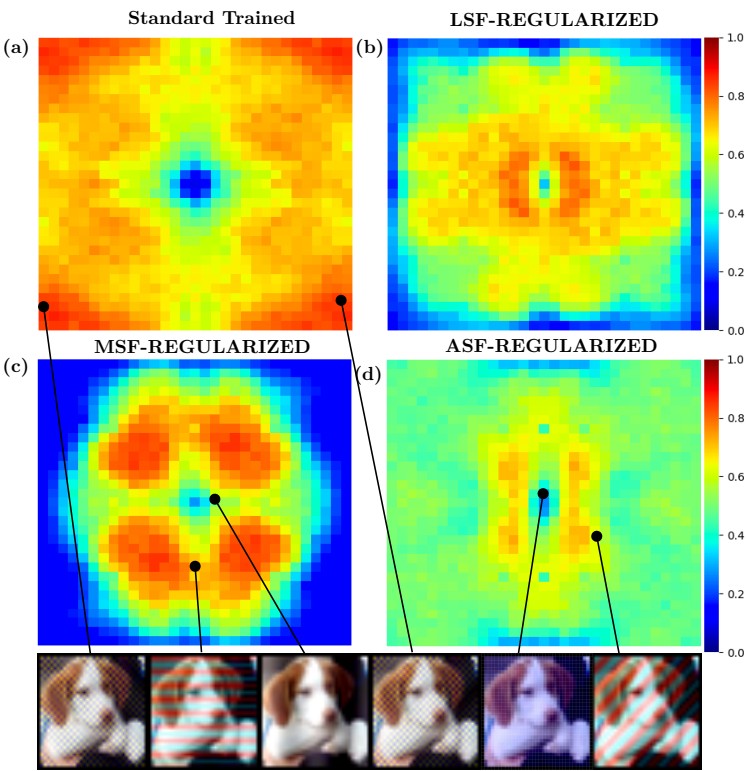

Figure 17: (CIFAR10) Heat map of error rates for each Fourier-mode corruption (low-frequencies close to the center). Each pixel in the heat map is the error of the model when the corresponding Fourier-mode noise ($\epsilon = 4$) is added to the inputs. The bottom row displays example images containing the corresponding Fourier-noise.

## E.2 SVHN

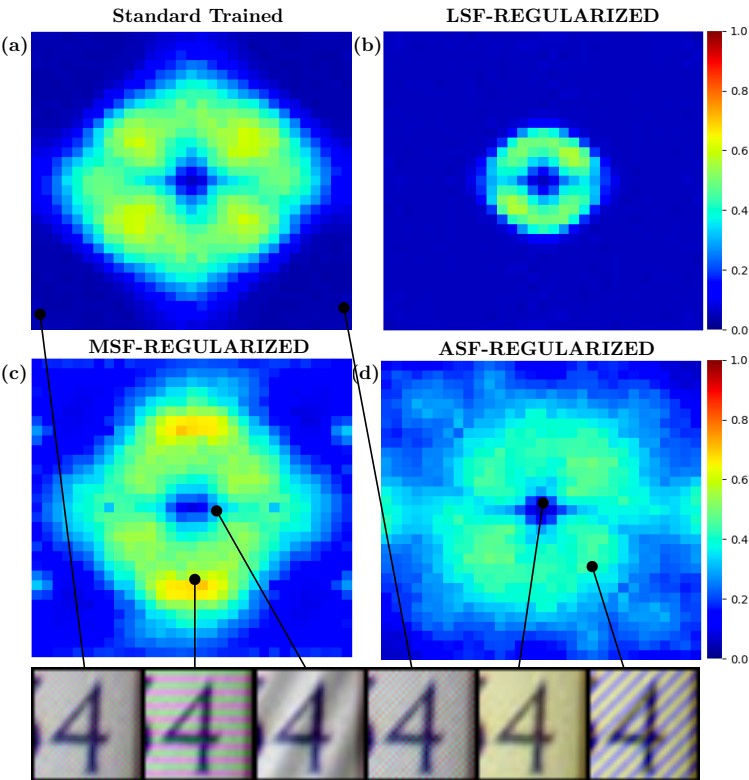

Figure 18: (SVHN) Heat map of error rates for each Fourier-mode corruption (low-frequencies close to the center). Each pixel in the heat map is the error of the model when the corresponding Fourier-mode noise ($\epsilon = 4$) is added to the inputs. The bottom row displays example images containing the corresponding Fourier-corruptions.

### E.3 Accuracy on Fourier-noise distortions

Table 6: Mean accuracy across all Fourier-noise corruptions averaged across 1024 randomly selected test samples for each corruption. $\ell_2$ norms of the additive Fourier-noise are $\epsilon \in \{3, 4\}$.

| Method | SVHN | | | CIFAR10 | | | CIFAR100 | | |
|---|---|---|---|---|---|---|---|---|---|
| | clean | $\epsilon{=}3$ | $\epsilon{=}4$ | clean | $\epsilon{=}3$ | $\epsilon{=}4$ | clean | $\epsilon{=}3$ | $\epsilon{=}4$ |
| Std. Train | 96.4 | 81.9 | 77.4 | 94.9 | 40.8 | 31.5 | 76.2 | 22.3 | 14.9 |
| LSF–REGULARIZED | 95.1 | **92.1** | **91.0** | 87.1 | 52.4 | 47.5 | 62.5 | 33.9 | 30.0 |
| MSF–REGULARIZED | 93.1 | 77.1 | 70.9 | 90.6 | **62.4** | **54.3** | 70.7 | **42.6** | **37.3** |
| ASF–REGULARIZED | 96.4 | 78.3 | 71.1 | 87.9 | 60.8 | 48.7 | 67.0 | 21.6 | 14.6 |

## F    Patch-shuffling images

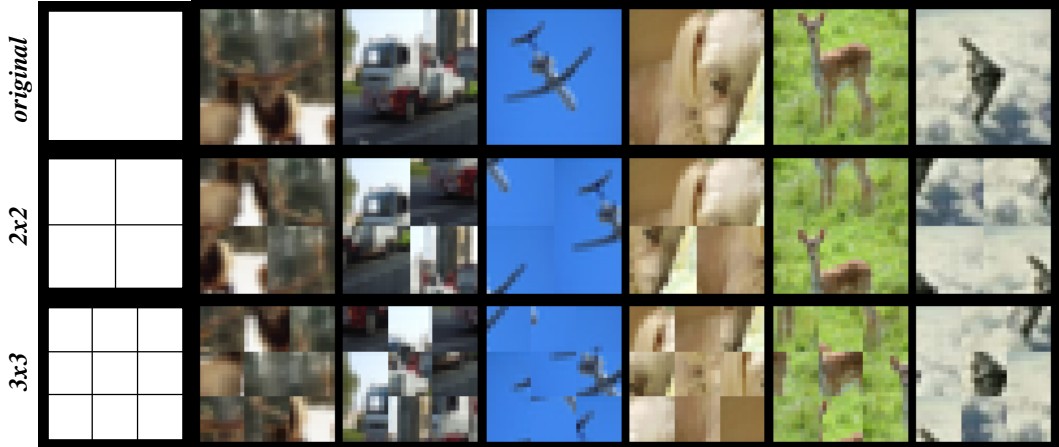

Figure 19: Patch-shuffling: Images are partitioned into squares whose positions are randomly exchanged. This operation destroys global structure in the image and is used to evaluate the extent to which a model relies on global information.

## G   Sensitivity to $\lambda_{\text{SFS}}$

Table 7: Clean accuracy on CIFAR10 (ResNet50) at different $\lambda_{\text{SFS}}$ values

| Method | $\lambda_{\text{SFS}} = 0$ | $\lambda_{\text{SFS}} = 0.1$ | $\lambda_{\text{SFS}} = 0.2$ | $\lambda_{\text{SFS}} = 0.5$ | $\lambda_{\text{SFS}} = 1$ |
|---|---|---|---|---|---|
| LSF-REGULARIZED | 94.8 | 93.8 | 91.1 | 87.1 | 84.6 |

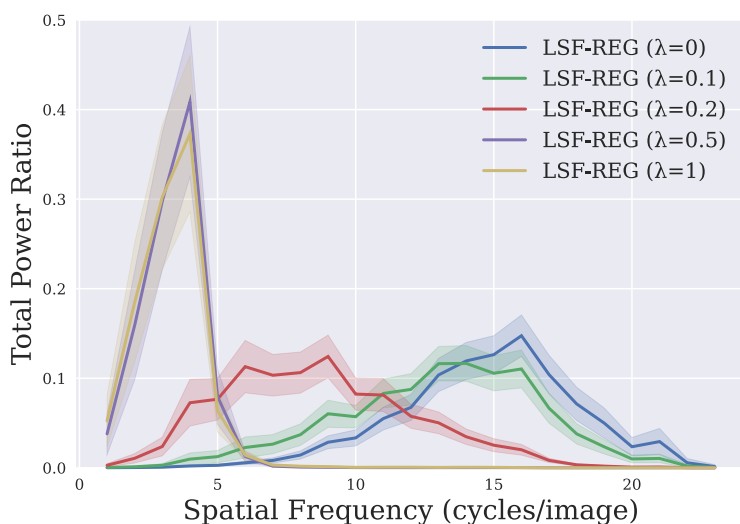

Figure 20: Fourier-sensitivity of LSF-REGULARIZATION at different $\lambda_{\text{SFS}}$ on CIFAR10 (ResNet50).

