# OpenReview forum: "Fourier Sensitivity and Regularization of Computer Vision Models"
_TMLR — Accepted by TMLR_

### Review · Reviewer_h6JT · 2022-09-01

**Summary Of Contributions:**

The paper presents an approach to compute 'Fourier sensitivity' which is a measure of what frequency bands the model is sensitive to. Then, they discuss how this can be used to regularise the model while training to control the frequency bias of the model. Such regularised models show improved robustness to adversarial noise as well as OOD examples. Experiments are performed on popular image classification datasets with various architectures.

**Broader Impact Concerns:**

The paper is of theoretical nature where it provides a tool to inspect model robustness. It may be possible to use this tool with malicious intent.

**Requested Changes:**

1. Eq.1 is not clear. What does the curly bracket represent? It is better to improve clarity.
2. Section 3.4 table, what is $\tilde{P}_k$?


**Strengths And Weaknesses:**

## Strengths
1. Fourier sensitivity is a useful quantity to understand robustness properties of a model. The approach to compute it is neat and easy to adopt as it just depends on the Jacobian of the network.
2. The experiments show Fourier sensitivity of different models and how this quantity is affected in adversarial training and training with Gaussian noise augmented images. Additionally, the Fourier regularisation shows that the 'sensitive' frequency band can be controlled while training and it yields improved robustness.
3. The experiments backed discussion in Section 4 is valuable. Overall the paper is well written.

## Weaknesses
I didn't notice any major weaknesses of the paper. One writing suggestion is to always define the symbols used in the equations and captions. It is hard to dig them from the text.

---

> ### Author Response · Authors · 2022-09-30
> **Revision**
>
> We thank the reviewer for their constructive comments and questions about our work. We have responded to the requested changes below. Edits in the paper were made in blue font.
>
> **‘Eq.1 is not clear. What does the curly bracket represent? It is better to improve clarity.’**
>
> Thanks for raising this point. In Eq. 1, the curly brackets represent the *set* of the proportion of total power in each frequency band. To improve clarity, we have changed the curly brackets to square brackets to represent Fourier-sensitivity as a vector, i.e., $ f_{SFS}(x,y) = [ P_{1}, \dots, P_{N/\sqrt{2}} ]$.
>
> **‘Section 3.4 table, what is $\tilde{P}_k$?’**
>
> For ASF-regularization, we exclude very high frequency terms ($k>N/2$), which is reflected in the $\tilde{P}_k$ terms. To improve clarity, we repeated the definition of $\tilde{P}_k$ from Section 3.1 (Notation) in Section 3.4, i.e., the proportion of power in frequency bands within the largest circle inscribed in the power-matrix, P.

---

> > ### Comment · Reviewer_h6JT · 2022-10-10
> > **Acknowledgement**
> >
> > Thanks for addressing the comments.

---

### Review · Reviewer_14vs · 2022-09-19

**Summary Of Contributions:**

This work carries out an empirical study of the sensitivity of various deep vision models to perturbations in the Fourier domain of input images, where perturbations are divided by band. The measure of Fourier sensitivity used is the Fourier transform of input gradients. Regularizers towards this sensitivity proposed. A number of empirical studies are conducted. It is found the sensitivity of models varies by dataset, architecture, and training. The proposed regularizers are empirically shown to mitigate this sensitivity in select cases.


**Broader Impact Concerns:**

I did not find a broader impact statement included, however I do not harbor any concerns about the broader impacts of this work.


**Requested Changes:**

The exclusion of high-frequency perturbations (HSF) in this work appears to me to be an omission. The authors note in Section 5.2: *"as the baseline models for these datasets already have high frequency sensitives..., we did not train HSF regularized models"*. I am left wondering about the high frequency sensitivity and regularization results for Figure 6 and Tables 1,2,3,4. For completeness I would still like to see these experiments conducted. This point is important to securing my recommendation.

I suggest that the authors drop claims of the neuroscience connection as fact. I don't think this is an appropriate venue to try to justify that claim. I am in favor of including the discussion of this connection in Appendix but with softened claims.

I have doubts that the "basis trick" proposed by the authors should be a theorem. I found that the presentation as such reduced the clarity of the paper, and suggest presenting it as an explicit calculation. This point would strengthen the clarity of the work in my opinion.

Why weren't the results in Figure 5c replicated for the models in Figure 5a? I suspect this is because of computational cost, however it would improve the completeness of the paper to include them.



**Strengths And Weaknesses:**

**Strengths**

Fourier analysis is fundamental in the theory of functions. Therefore it is a natural choice to use these techniques in the study of deep vision models. A careful and complete empirical study is a welcome contribution.

The empirical results in this paper appear to me well-founded. No theoretical explanations are given, but this seems fine to leave for future work.

Figure 4 convinces me that adversarially trained models are more sensitive to low frequency perturbations than standardly trained models, although only for CIFAR10 and Imagenet. It remains open why this is not the case for ~CIFAR-10~ SVHN, or what property of the data exactly leads to this sensitivity.

Figure 5 convinces me that there is indeed a variable Fourier sensitivity across models, e.g.  InceptionV3 models are more sensitive to low frequencies, but Vision Transformer models are more sensitive to middle and high frequencies. Why this occurs is still an open question.

Figure 6 is convinces me that Fourier-regularization has the intended effect of enhancing sensitivity towards a given frequency band.

Patch shuffling is an appropriate empirical test of low-frequency bias.

The effect of Fourier-regularization on select OOD shifts is interesting (Table 4). It suggests that specific data shifts may be partially addressed by the appropriate Fourier regularization.

**Weaknesses**

This is primarily an empirical work. It would of course be desirable to have theoretical explanation of the observations made here, but I am okay with the authors leaving this for future work.

I don't find the "basis trick" proposed to be a significant theoretical contribution: it is simply the application of the chain rule for multivariate functions. I do think it is an appropriate measure for the empirical investigation conducted.

A possible connection to the neuroscience of spatial frequency processing in the brain is made in the intro and elaborated in Appendix A. This is potentially interesting, but I found the wording in the intro to be overly strong. It suggests that this connection is indeed already established fact: (*"In fact, spatial frequencies are differentially processed..."*). I cannot accept this claim of fact without further justification. Moreover these claims already appear soften in Appendix A (*"it has been posited..."*).

I am unclear on why the authors use the term "Jacobian" instead of gradient in this work. I cannot detect any reason why the term Jacobians should be used. All model functions in this work have scalar outputs, for which the term gradient is usually used. Perhaps the authors can clarify.

---

> ### Author Response · Authors · 2022-09-30
> **Revision**
>
> We thank the reviewer for their constructive comments and questions about our work. We have responded to the concerns below. Edits in the paper were made in blue font.
>
> **I am unclear on why the authors use the term "Jacobian" instead of gradient in this work. I cannot detect any reason why the term Jacobians should be used. All model functions in this work have scalar outputs, for which the term gradient is usually used. Perhaps the authors can clarify**
>
> Thank you for raising this point. We agree that the term 'Jacobian' is commonly used for vector-valued functions, and 'gradient' is used for scalar-valued functions. We used the term 'Jacobian' as it is the generalization of the gradient. To improve clarity, we have replaced 'input-Jacobian' with 'input-gradient' in the paper.
>
> **The exclusion of high-frequency perturbations (HSF) in this work appears to me to be an omission. The authors note in Section 5.2: "as the baseline models for these datasets already have high frequency sensitives..., we did not train HSF regularized models". I am left wondering about the high frequency sensitivity and regularization results for Figure 6 and Tables 1,2,3,4. For completeness I would still like to see these experiments conducted. This point is important to securing my recommendation**
>
> As requested, we have added HSF-regularization to Section 3.4 (Fourier-regularization) and updated Figure 6 and Tables 1,2,3,4. The HSF-regularized model has high clean accuracy (Table 2), is less reliant on global image structure, as expected (Table 1), has comparable or more robustness to Fourier-filtering than the standard trained model (Tables 2,3). It was more robust to weather and digital corruptions compared to blurring, which expectedly requires low-frequency bias (Table 4).
>
> **I suggest that the authors drop claims of the neuroscience connection as fact. I don't think this is an appropriate venue to try to justify that claim. I am in favor of including the discussion of this connection in Appendix but with softened claims**
>
> Thank you for raising this concern. Based on our reading of the relevant literature, we found evidence published over the decades for different spatial-frequency centers in the visual cortex, which we have briefly summarized in Appendix A. However, as we are not neuroscience experts, we are not able to make conclusive statements about this subject. Hence, as suggested, we have toned down the statement in the introduction to “Moreover, spatial frequencies may also be differentially processed in the brain” instead of “In fact, spatial frequencies are differentially processed in the brain”.
>
> **I have doubts that the "basis trick" proposed by the authors should be a theorem. I found that the presentation as such reduced the clarity of the paper, and suggest presenting it as an explicit calculation. This point would strengthen the clarity of the work in my opinion**
>
> Thanks for the suggestion. To improve clarity, we have re-arranged Section 3.2 (Basis Trick) to first describe the basis trick as an explicit computation *before* the formal statements. Furthermore, taking into consideration the comment, we have renamed Theorem 1 to Proposition 1 while retaining it for its generality. We hope this improves the exposition in this section.
>
> **Why weren't the results in Figure 5c replicated for the models in Figure 5a? I suspect this is because of computational cost, however it would improve the completeness of the paper to include them**
>
> Thanks for raising this point. We have added adversarially trained ResNet18 to Figure 5c, whose Fourier-sensitivity is consistent with the other architectures in the figure. As adversarial training is significantly more computationally expensive than standard training, we were not able to train adversarially-robust vision transformers and other architectures on ImageNet in this limited time-period due to resource constraints. We leave this to future work.

---

> > ### Comment · Reviewer_14vs · 2022-10-06
> > **Acknowledgement**
> >
> > Thanks to the authors for addressing my concerns.

---

### Review · Reviewer_dm8R · 2022-09-26

**Summary Of Contributions:**

This paper focuses on the relationship between the Fourier spectrum of the images and the sensitivity of the contemporary neural networks. The paper proposes a way to measure the Fourier sensitivity of such models and then conducts a number of experiments to showcase the Fourier sensitivity of different models (e.g. Resnet-based, Transformers, etc). The paper also indicates what is the relationship of this sensitivity with adversarially-trained models.

**Broader Impact Concerns:**

There is no broader impact statement at the moment.

**Requested Changes:**

* There are no limitations of the study at the moment.

* The writing could be refined. Is the "spectral bias of neural networks" (sec. 3.4) referring to the spectral bias observed in the literature (e.g. in [1-4])? Because I found no reference of this related work, and I think the analysis here is based on the Fourier analysis of the data not on the neural network level.

* Is the $\epsilon=3$ used in Fig. 5 standard for PGD-L2 attacks?



[1] On the Spectral Bias of Neural Networks, ICML'19.

[2] The Spectral Bias of Polynomial Neural Networks, ICLR'22.

[3] On the Spectral Bias of Convolutional Neural Tangent and Gaussian Process Kernels, NeurIPS'22.

[4] Towards Understanding the Spectral Bias of Deep Learning.

**Strengths And Weaknesses:**

[+] Understanding the sensitivity of the models, even through experimental validation, is significant.

[+] The paper conducts a number of experiments, which validates different aspects of the sensitivity, e.g. different architectures, datasets.

[-] The novelty in the technical part is not very clear. The relationship of different frequencies (of the data) and the model sensitivity has been observed numerous times in the past.

[-] The writing could be further polished (see comment below).

* Even though this is not required for publication, it would be recommended to make the source code public.

---

> ### Author Response · Authors · 2022-09-30
> **Revision**
>
> We thank the reviewer for their constructive comments and questions about our work. We have responded to each concern below. Edits in the paper were made in blue font.
>
> **The novelty in the technical part is not very clear. The relationship of different frequencies (of the data) and the model sensitivity has been observed numerous times in the past.**
>
> We agree that the sensitivity of models to input frequencies has been empirically investigated using frequency-based interventions, as discussed in the Related Work (Section 2.1). Our novel contributions are as follows. First, we show that the Fourier-transform of the input-Jacobian provides the analytical sensitivity of the model to input frequencies. Second, using this principled definition, we studied Fourier-sensitivity across datasets, architectures and training methods. Third, we proposed Fourier-regularization based on the definition of Fourier-sensitivity, which is a principled approach to modify the frequency sensitivity of a model.
>
> **Even though this is not required for publication, it would be recommended to make the source code public**
>
> We will include a link to the source code on GitHub in the final version of the paper.
>
> **Is the "spectral bias of neural networks" (sec. 3.4) referring to the spectral bias observed in the literature (e.g. in [1-4])? Because I found no reference of this related work, and I think the analysis here is based on the Fourier analysis of the data not on the neural network level**
>
> Thanks for raising this point. We agree there may be confusion due to our use of the term "spectral bias", which has already been used in other work in a different context. Our study is at the input-level instead of the function-level as discussed in the linked papers. To improve clarity, we have changed all uses of the term “spectral bias” to “frequency sensitivity” in the paper.
>
> **Is the $\epsilon=3$ used in Fig. 5 standard for PGD-L2 attacks?**
>
> The L2-attack ($\epsilon=3$) is one of the commonly used threat models in robustness studies, e.g. see [1], [2], [3].
>
> [1] Does Robustness on ImageNet Transfer to Downstream Tasks? CVPR '22
>
> [2] Do Adversarially Robust ImageNet Models Transfer Better? NeurIPS ‘20
>
> [3] Certified Adversarial Robustness via Randomized Smoothing. ICML '19

---

> > ### Comment · Reviewer_dm8R · 2022-10-05
> > **Additional question**
> >
> > Dear authors, thank you for providing responses to my questions.
> >
> > I have an additional question on the empirical front: The paper mentions that "Vision transformers display sensitivity to mid-range as well as high frequencies". However, I have not understood which model this is referring to and the corresponding figures that illustrate this. Given that transformers are an increasingly important module, I think it is important to examine this.

---

> > > ### Author Response · Authors · 2022-10-05
> > > **Response**
> > >
> > > Thank you for the question. This refers to the Fourier-sensitivity of Vision Transformer (ViT) [1], visualized in Figure 5a (the green curve). We can observe two peaks in this curve, one in the mid-frequency range ($50<k<80$) and another in the high-frequency range ($120<k<160$). To improve clarity, we have added additional references to the figure and [1] in the paper.
> > >
> > > [1] An Image is Worth 16x16 Words: Transformers for Image Recognition at Scale, ICLR '21

---

> > > > ### Comment · Reviewer_dm8R · 2022-10-10
> > > > **Thanks for the clarification**
> > > >
> > > > I am thankful to the authors for the clarification.

---

### Decision · Action_Editors · 2022-11-27

**Recommendation:** Accept as is

**Comment:**

The submission addresses Fourier sensitivity of neural network models, and explores effects of model architecture, as well as the relationship to adversarial examples.  These topics are well addressed with experiments appropriately supporting the claims in the paper.  They are also topics of high interest in computer vision and neural network research.  The reviewers were unanimous in their opinion that the paper should be accepted without further revision.

**Audience:**

The topic is well positioned for the readership of TMLR, with the subject matter being an appropriate combination of computer vision, statistical learning, and signal processing.

**Claims And Evidence:**

The submission has unanimously been recommended for acceptance by the reviewers with no additional requested changes.  It is therefore suitable for publication in TMLR.